# NS1 Protein N-Linked Glycosylation Site Affects the Virulence and Pathogenesis of Dengue Virus

**DOI:** 10.3390/vaccines11050959

**Published:** 2023-05-08

**Authors:** Enyue Fang, Miao Li, Xiaohui Liu, Kongxin Hu, Lijuan Liu, Zelun Zhang, Xingxing Li, Qinhua Peng, Yuhua Li

**Affiliations:** 1Institute of Health Inspection and Quarantine, Chinese Academy of Inspection and Quarantine, Beijing 100176, China; dlu_fangenyue@163.com (E.F.); kongxinhu@sina.com (K.H.); lljyhxx@163.com (L.L.); 2Department of Arbovirus Vaccine, National Institutes for Food and Drug Control, Beijing 102629, China; limiao1711@163.com (M.L.); liux1aohui@163.com (X.L.); zelun128@hotmail.com (Z.Z.); lxx0320@outlook.com (X.L.); pengqinhua666@163.com (Q.P.); 3Vaccines R&D Department, Changchun Institute of Biological Products Co., Ltd., Changchun 130000, China

**Keywords:** dengue virus, N-linked glycosylation, neurovirulence, chimeric vaccine

## Abstract

Live attenuated vaccine is one of the most effective vaccines against flavivirus. Recently, site-directed mutation of the flavivirus genome using reverse genetics techniques has been used for the rapid development of attenuated vaccines. However, this technique relies on basic research of critical virulence loci of the virus. To screen the attenuated sites in dengue virus, a total of eleven dengue virus type four mutant strains with deletion of N-glycosylation sites in the NS1 protein were designed and constructed. Ten of them (except for the N207-del mutant strain) were successfully rescued. Out of the ten strains, one mutant strain (N130del+207-209QQA) was found to have significantly reduced virulence through neurovirulence assay in suckling mice, but was genetically unstable. Further purification using the plaque purification assay yielded a genetically stable attenuated strain #11-puri9 with mutations of K129T, N130K, N207Q, and T209A in the NS1 protein and E99D in the NS2A protein. Identifying the virulence loci by constructing revertant mutant and chimeric viruses revealed that five amino acid adaptive mutations in the dengue virus type four non-structural proteins NS1 and NS2A dramatically affected its neurovirulence and could be used in constructing attenuated dengue chimeric viruses. Our study is the first to obtain an attenuated dengue virus strain through the deletion of amino acid residues at the N-glycosylation site, providing a theoretical basis for understanding the pathogenesis of the dengue virus and developing its live attenuated vaccines.

## 1. Introduction

Dengue fever is a mosquito-borne viral infection caused by the dengue virus (DENV), which is widely spread in tropical and subtropical regions. The virus is transmitted in the “human-*Aedes aegypti*-human cycle” through the bite of DENV-infected *A. aegypti* mosquitoes. The sources of infection for dengue fever are primarily female *A. aegypti* mosquitoes and, to a lesser extent, dengue fever patients, latent infections, and non-human primates infected with dengue virus [1]. According to the World Health Organization (WHO), dengue cases increased more than eight times in the last 20 years, from 505,430 cases in 2000 to more than 2.4 million cases in 2010 and 5.2 million cases in 2019 [2]. Currently, dengue fever is still aggravating worldwide, with the epidemic area dominated by the co-prevalence of multiple virus serotypes, and which has become one of the acute infectious diseases of global concern.

According to the serum neutralization test and the antigenicity of the envelope protein (E), DENV is primarily divided into four serotypes. The fifth serotype (DENV-5) was identified in a patient’s blood in Sarawak, Malaysia, in 2007, but it needs to be confirmed with more data [3]. Dengue virus consists of three structural proteins (C, prM/M, and E) and seven non-structural proteins (NS1-5). The structural proteins are the main target antigens for vaccine design, while the non-structural proteins are mainly involved in RNA replication and viral packaging. NS1 protein is a glycoprotein possessing multiple forms [4,5,6] and is considered a protective antigen in dengue vaccines [7]. The immune response generated against NS1 protein can effectively prevent severe dengue fever. Dengue vaccines have been developed for nearly a century, beginning in the 1920s. However, most vaccines obtained at the early stage using viral inactivation or attenuation via successive passaging of the infected cells were ineffective. Later on, scholars around the world successfully developed live attenuated virus vaccines using reverse genetics, including Dengvaxia, a tetravalent YFV/DENV chimeric vaccine by Pasteur [8,9,10], tetravalent DENVax by Takeda [11,12,13,14,15], and tetravalent TV003/TV005 by the National Institute of Allergy and Infectious Diseases [16,17,18,19,20,21,22,23].

N-glycosylation is most common in organisms and refers to the covalent attachment of sugar chains to asparagine residues in the conserved motif NxS/T, with x denoting an amino acid other than proline [24]. During the infection cycle of the dengue virus, the N-glycosylation modifications on the E and NS1 proteins play important roles in the adsorption, invasion, maturation, assembly, and secretion of the virus [25,26]. N-glycosylation modifications not only hide the neutralizing epitopes on the surface of viral proteins, but also participate in viral infection and affect virulence [27]. Two N-glycosylation sites, N130 and N207, are present in the dengue virus NS1 protein and are highly conserved among flaviviruses. Deleting all N-glycosylation sites in the NS1 protein of West Nile virus (WNV) results in complete loss of neurovirulence of the attenuated virus to mice, but retains promising immunogenicity [28]. In this study, we constructed multiple NS1 protein N-glycosylation site deletion mutant strains using the reverse genetics technique and screened a dengue mutant strain with attenuated virulence and genetic stability for the development of attenuated dengue virus vectors and dengue chimeric vaccines.

## 2. Materials and Methods

### 2.1. Viruses, Plasmids, Cells, and Animals

DENV-4 Ban18 strain, which was isolated in 1981 from Xishuangbanna, Yunnan, China, and DENV-4 Ban18HK20 strain, which was obtained by passing DENV-4 Ban18 strain on primary hamster kidney cells for 20 generations, were from the Division of Arboviral Vaccines, National Institutes for Food and Drug Control (NIFDC), China. The infectious cloning plasmid pSPTM-DENV of DENV-4 Ban18HK20 strain, the chimeric dengue virus subcloning plasmid pUC57-DENV4/1-A, and the dengue chimeric virus rDENV4/1 were from NIFDC. The specific pathogen-free 3-day-old BALB/c suckling mice and 4-day-old ICR suckling mice were supplied by the Center of Animal Breeding, NIFDC. Vero cells were from ATCC and preserved in the Division of Arboviral Vaccines for passaging.

### 2.2. Design of Dengue Virus NS1 Protein N-Glycosylation Site Mutations

It has been reported that NS1 glycosylation affects WNV pathogenicity, and deletion of glycosylation sites significantly reduces WNV virulence and can be used as a design strategy for live attenuated vaccines [28]. Therefore, we constructed 11 glycosylation mutations of the dengue virus NS1 protein using its infectious cloning of Ban18HK20 strain, including eight single-site mutations and three double-site combined deletion mutations (Figure 1). These mutations included (i) a mutation of amino acid residue 130 from N to A (N130A), (ii) a mutation of amino acid residue 207 from N to A (N207A); (iii) a mutation of amino acid residue 130 from N to Q (N130Q); (iv) a mutation of amino acid residue 207 from N to Q (N207Q); (v) mutations of amino acid residues 130, 131, and 132 from NST to QQA (130-132QQA); (vi) mutations of amino acid residues 207, 208, and 209 from NQT to QQA (207-209QQA); (vii) a deletion of amino acid residue 130 (N130-del); (viii) a deletion of amino acid residue 207 (N207-del); (ix) N130A combined with N207A mutation (N130A+N207A); (x) 130-132QQA combined with 207-209QQA mutation (130-132QQA+207-209QQA); and (xi) N130-del combined with 207-209QQA mutation (N130-del+207-209QQA).

### 2.3. Construction Strategy for Molecular Cloning

To construct infectious clones of the N-glycosylation site mutant strains of dengue virus NS1 protein, we selected fragments flanking the NS1 protein glycosylation sites with suitable restriction sites at the 5′ and 3′ ends, PCR-amplified them, and cloned them using In-Fusion cloning into a dengue virus infectious clone plasmid pSPTM-DENV (Ban18HK20) reported previously [29,30]. Appendix A shows the amplification primers and enzymatic sites for cloning. Briefly, the plasmid was cleaved using the corresponding restriction endonucleases listed in Appendix A and separated by agarose gel electrophoresis. The target fragment was recovered from agarose gel and purified. Two fragments containing the mutation site were amplified using two pairs of primers with homologous arms that bind complementarily to the end of the linearized vector. After removal of the plasmid template using *Dpn*I (Cat. R0176L; NEB), the PCR products were purified using a DNA purification kit (Cat.9761; TaKaRa, Kusatsu, Shiga). Homologous recombination was performed by mixing the two PCR amplification fragments with a linearization vector in a 2:2:1 molar ratio in 5× In-Fusion HD Enzyme Premix (Cat. 639650; TaKaRa). The recombinant products were transformed into DH10B competent cells, and single clones were selected for sequencing and identification. The correct plasmids were extracted and used for in vitro transcription.

### 2.4. In Vitro Transcription, Transfection, and Virus Rescue

The infectious clone plasmids were linearized using *Xho*I, which cleavage site had been artificially added to the 3′ end of the viral cDNA and digested using mung bean nuclease to make the sticky ends. After inactivating mung bean nuclease by adding SDS at a final concentration of 2%, the DNA fragments were purified and recovered using the V-Elute Gel Mini Purification Kit (Cat. ZPV202; Beijing Zoman Biotechnology Co., Ltd., Beijing, China) and used as a template for in vitro transcription with a RiboMAX Large Scale RNA Production System (Cat. P1280; Promega, Madison, WI, USA). The 5′ end of viral RNA was capped using a Ribo m7G Cap Analog (Cat. P1712; Promega) to mimic the native structure of the virus. After removal of the DNA template using DNase I, RNAs were purified using the RNeasy MinElute Cleanup Kit (Cat. 74204; QIAGEN, Hilden, Germany) and transfected into Vero cells using a Gene Pulser Xcell Electroporation System using a Gene Pulser Electroporation Buffer (Cat. 1652676; Bio-Rad, Hercules, CA, USA) with the voltage set at 220 V, capacitor set at 300 μF, cuvette gap at 0.4 cm, and resistor at none. Significant cytopathic effects were observed 5–7 days after electro-transfection, and cell supernatants were harvested as rescue viruses and stored frozen at −80 °C.

### 2.5. Virus Titer Assay

Vero cells were seeded in six-well plates at 1 × 10^6^ cells/well. The supernatant was discarded at cell confluence of 80–90%, and cells were infected with the virus at 10-fold serial dilution (10^−1^–10^−6^) for 1 h at 37 °C and 5% CO_2_. The viruses were discarded, and cells were overlaid with methylcellulose and incubated for 7 days. After removing the overlay, cells were stained with crystalline violet for 30 min and washed. The plates were air-dried, and the number of plaques was counted. The virus titer was expressed as the log of the number of plaque-forming units (PFU) per mL or log_10_ (PFU/mL).

### 2.6. Western Blotting

Vero cells were infected with viruses at a multiplicity of infection (MOI) of 1 and incubated at 37 °C with 5% CO_2_ for 48 h. After washing the cells with PBS, cells were lysed with RIPA lysis buffer (Sigma, Cat.R0278-50ML, St. Louis, MO, USA) on ice for 30 min. Cell lysates were collected using centrifugation at 12,000× *g* for 30 min at 4 °C and used to detect protein concentrations. The protein samples were mixed with NuPAGE 4× Loading Buffer (Invitrogen, Cat.NP0007, Waltham, MA, USA), denatured at 70 °C for 10 min, subjected to SDS-PAGE electrophoresis, and transferred onto PVDF membranes using the iBlot 2 Gel Transfer Device (Thermo Fisher Scientific, IB21001, Waltham, MA, USA). After PVDF membranes were blocked with 5% BSA for 2 h at room temperature, the membranes containing target proteins were incubated with a 4G2 antibody (Novus Biologicals, Cat.NBP2-52709-0.2mg, Centennial, CO, USA) or NS1 antibody (Arigo Biolaboratories, Cat.ARG65660, Taiwan, China), respectively, and the membranes containing the internal reference protein were incubated with a GAPDH antibody (TransGen Biotech Co., Ltd., HC301, Beijing, China) overnight at 4 °C on a shaker. After washing the membrane with PBST, the membranes were further incubated with HRP-conjugated goat anti-mouse IgG secondary antibody (TransGen Biotech Co., Ltd., HS201, Beijing, China) for 1.5 h at room temperature before being washed with PBST. The signals were detected and imaged with a chemiluminescent imaging system.

### 2.7. Indirect Immunofluorescence Assays

Vero cells were seeded in a 96-well plate, infected with the virus at an MOI of 1, and cultured at 37 °C in an incubator with 5% CO_2_ for 48 h. After discarding the supernatants, cells were washed once with PBS and fixed using pre-cooled 80% acetone solution for 30 min at 4 °C. After that, acetone was removed and cells were air-dried at room temperature, washed once with PBS, and incubated with a primary anti-Dengue virus E glycoprotein antibody (Abcam, ab41349, Cambridge, UK) for 30 min at room temperature. After being washed three times with PBS, cells were incubated with fluorescent secondary goat anti-mouse IgG H&L (Abcam, ab150113) for 30 min at room temperature in the dark, washed three times with PBS, and counter stained with DAPI to visualize the nuclei for 5 min at room temperature in the dark. After being washed three times with PBS, cells were observed under an inversion fluorescence microscope and photographed for recording.

### 2.8. Virus Genome Sequencing and Genetic Stability Assay

RNA was extracted from the rescue virus using a QIAamp Viral RNA Mini Kit (QIAGEN, Cat. 52904) and reverse transcribed into cDNA using a GoScript™ Reverse Transcription System (Promega, Cat. A5000). Primers were designed to amplify the full length of the viral genome in segments, and the amplification products were sent to Sangon Biotech Co., Ltd. (Shanghai, China) for Sanger sequencing. The walking sequencing primers for amplification products of the E and NS1 regions of the dengue virus genome are listed in Appendix A. For genetic stability analysis, viruses were passed through 5 generations in Vero cells, and viral RNA was extracted from cell supernatants of the 1st, 3rd, and 5th generations and subjected to RT-PCR sequencing to identify the mutation sites.

### 2.9. Mouse Experiments

Specific pathogen-free 3-day-old BALB/c suckling mice and 4-day-old ICR suckling mice were selected for neurovirulence assays of the dengue virus. In brief, mice were intracerebrally injected with 0.02 mL of virus diluted with PBS containing 2% FBS and observed for 21 days. The average survival time (AST) and mortality of mice were calculated. The neurovirulence of different mutant viruses in suckling mice was observed, and a half-lethal dose (LD_50_) was calculated based on the Reed-Muench method.

### 2.10. Virus Plaque Purification Assay

The virus to be purified was serially diluted 10-fold and used to infect Vero cells in a six-well plate at 37 °C in an incubator with 5% CO_2_ for 1 h. After the virus was discarded, cells were overlaid with a layer of agar overlay and incubated for 5 days, followed by overlaying the second layer of agar containing neutral red staining solution and incubating for 24 h. The monoclonal viral strain was isolated under the light at seven days post-infection and cultivated in monolayer Vero cells in a six-well plate, followed by passaging in T25 cell flasks for expansion.

### 2.11. Construction of Dengue Chimeric Virus

The chimeric plasmid pSPTM-DENV4/1 was constructed by replacing the corresponding gene in the infectious cloning plasmid pSPTM-DENV of DENV type 4 with the prM-E gene of DENV type 1 (79–116 strain). Since the full-length infectious clone of DENV4/1 was unstable, it was cloned via segmentation followed by in vitro ligation (Figure 2). First, pUC57-DENV4/1-A with N207Q+T209A and K129T+N130K+N207Q+T209A mutations and pSPTM-DENV(mut) with NS2A-E99D mutation were constructed. Second, the pUC57-DENV4/1-A was digested with *Asc*I and *Afl*II to generate a 3.4 kb linear fragment DENV4/1-A, and pSPTM-DENV(mut) was digested with *Xho*I and *Afl*II to generate a 7.3 kb fragment DENV4/1-B. Third, the fragment DENV4/1-A and fragment DENV4/1-B were ligated overnight at 4 °C by T4 DNA ligase. The 10 kb fragment was purified after electrophoresis and used as the chimeric viral cDNA for in vitro transcription and virus rescue. 

### 2.12. Statistical Analysis

All statistical analyses were performed using GraphPad Prism 9 (GraphPad Software Inc., San Diego, CA, USA). The differences among multiple groups were analyzed using one-way analysis of variance (ANOVA) followed by Dunnett’s test. Survival analysis was performed using Log-rank (Mantel–Cox) test. A *p*-value less than 0.05 was regarded as statistically significant. * *p* ≤ 0.05, ** *p* ≤ 0.01, *** *p* ≤ 0.001, and **** *p* ≤ 0.0001.

## 3. Results

### 3.1. Identification of Rescue Viruses

Virus titer determination revealed that the N207-del strain did not show any detectable level of virus (no viral titer was detected). N130A, N207A, N130Q, N207Q, 130-132QQA, 207-209QQA, and N130-del strains showed higher levels of virus (viral titers above 5.0 log_10_ PFU/mL); the three strains with combined deletion of the two glycosylation sites showed similar levels of virus (All above 7.0 log_10_ PFU/mL). The size of plaques formed by the rescued viruses was not significantly different (Figure 3A).

Among the mutant strains with deletion of one glycosylation site, the expression of NS1 protein was significantly reduced in the N130-del strain. However, in the N207-del strain, both viral-specific E protein and NS1 protein expression were not detected, indicating that deletion of the N207 site may impair the rescue of the virus. By contrast, higher expression of virus-specific E protein and NS1 protein was detected in all three mutant strains with deletion of both glycosylation sites (N130A+N207A, 130-132QQA+207-209QQA, and N130-del+207-209QQA). Additionally, the molecular weights of the NS1 proteins of the glycosylation-deficient mutant viruses were found to be lower than those of the parental Ban18HK20 strain without glycosylation deletion (Figure 3B). Indirect immunofluorescence assays showed that only the N207-del strain failed to express virus-specific E protein (Figure 3C). This may indicate that deletion of the N207 site affects viral replication or packaging. RT-PCR amplification of the RNA of the rescue viruses showed that only N207-del was not amplified. Sequencing of viral genomes of the remaining ten mutant strains showed consistent results with the target sequences (Figure 3D).

### 3.2. Dengue Virus Possesses an Attenuated Phenotype after Deletion of the N-Glycosylation Site in the NS1 Protein

The NS1 protein of the dengue virus Ban18HK20 strain contains two potential N-glycosylation motifs (130-132NST and 207-209NQT), and mutation of either the first or third amino acid residues of these motifs results in N-glycosylation deletion mutations. This study first constructed eight mutant strains with a single N-glycosylation site deletion at position 130 or 207. Only seven were used for subsequent studies because the deletion of asparagine residue at position 207 failed to rescue the virus. The impact of these mutations on virulence was tested using neurovirulence assays in suckling mice, and mutations with attenuated phenotypes were further mutated to find further attenuated strains.

The mutant N130del+207-209QQA strain showed the most significantly attenuated neurovirulence compared with other mutant strains, as indicated by its greater LD_50_ and longer average survival time in 3-day-old BALB/c suckling mice. However, it presented higher mortality at a low dose of 2.1 log_10_ PFU than at a high dose of 3.1 log_10_ PFU (Table 1), suggesting that the strain is genetically unstable and may further mutate to other strains with different virulence. Thus, additional plaque purification is necessary to further confirm its true virulence. This strain also showed significantly attenuated neurovirulence in 4-day-old ICR suckling mice (Figure 4, Table 2). However, unlike in BALB/c suckling mice, the N130-del mutation also resulted in significantly attenuated neurovirulence in 4-day-old ICR suckling mice, suggesting that N130 may be a critical locus affecting viral neurovirulence.

### 3.3. Plaque Purification of the Mutant N130del+207-209QQA Strain and Screening of Attenuated Strains

The mutant N130del+207-209QQA strain with the lowest virulence was sequenced after five serial passages in Vero cells. The results showed that three mutations, K112N, K129T, and del130K, appeared in the NS1 protein starting from the third generation and were maintained in the fifth generation (Table 3). Since the mutant N130del+207-209QQA strain was genetically unstable and had higher virulence at low doses in suckling mice, plaque purification was performed to isolate genetically stable attenuated clonal strains. Ten purified clonal strains (puri1–10) were injected intracerebrally into the 4-day-old ICR suckling mice. The results showed that of the ten purified clonal strains, #11-puri7 and #11-puri9 were not lethal to the suckling mice even at a high dose of 100 PUF (Table 4), showing significantly attenuated characteristics.

Viral genome sequencing of the ten purified clonal strains (puri1–10) showed that #11-puri7 and #11-puri9 possessed the same mutation sites (K129T, N130K, N207Q, and T209A) in the NS1 protein, and E99D in the NS2A protein (Table 5), indicating that these mutations may affect virulence. 

### 3.4. Identification of Virulence Loci for Attenuated Mutants

The five amino acid mutation sites (NS1-K129T, N130K, N207Q, T209A, and NS2A-E99D) identified in this study were introduced into the dengue virus infectious clone plasmid, and their impacts on dengue virus virulence were evaluated. The plaque formation results showed that the rescue viruses with the same genotype as #11-puir9 showed significantly smaller plaques than the Ban18HK20 strain (Figure 5), consistent with previous reports that attenuated viruses had smaller plaques [31,32]. The neurovirulence test results showed that the rescue virus with the same genotype as #11-puir9 remained non-lethal to suckling mice after challenging at a dose of 100 PFU (Table 6), showing a significantly more attenuated profile than the Ban18HK20 strain. These results confirmed that K129T, N130K, N207Q, and T209A mutations in NS1 protein and E99D in NS2A protein could affect dengue virus virulence.

Since all mutations affecting virulence identified in our study were located in the non-structural proteins of dengue viruses, these mutations could be used for constructing attenuated dengue chimeric viruses. Because the DENV4/1 chimeric virus constructed in our previous study is still strongly neurovirulent to suckling mice, N207Q+T209A+E99D or K129T+N130K+N207Q+T209A+E99D mutations were introduced into the chimeric virus in this study. The plaque formation results showed that rDENV4/1 with N207Q+T209A+E99D mutations resulted in both large and small plaques, while rDENV4/1 with K129T+N130K+N207Q+T209A+E99D mutations resulted in only small plaques as #11-puri9, significantly different from non-mutated DENV4/1 (Figure 6). Neurovirulence results also demonstrated that the rDENV4/1 with N207Q+T209A+E99D mutations showed strong neurovirulence in the 4-day-old ICR suckling mice, similar to the non-mutated DENV4/1. By contrast, the rDENV4/1 with K129T+N130K+N207Q+T209A+E99D mutations was non-lethal to suckling mice and shared the same attenuated characteristics as the #11-puri9 strain (Table 7). These data suggest that K129T and N130K mutations significantly reduce neurovirulence in suckling mice.

### 3.5. Genetic Stability of Plaque-Purified Attenuated Dengue Virus Strains

The genetic stability of the #11-puri9 purified strain and the rescued virus of its infectious clone was confirmed via passaging to passage 5 (P5) in Vero cells and sequencing the viral genome of the third and fifth generations. The results showed that the #11-puri9 strain was genetically stable with no nucleotide mutations (Table 8). The adaptive mutation of the attenuated locus identified in the #11-puri9 strain (Figure 7) could be used in constructing attenuated dengue viruses or chimeric dengue viruses with other serotypes.

## 4. Discussion

This study found that the deletion of amino acids in the NS1 protein of the dengue virus affected its packaging and stability, leading to a lack of detectable viral RNA and proteins during rescue and after multiple passages. Our results support previous findings that the glycosylation modification of NS1 protein in dengue and West Nile viruses can impact virus secretion and stability [33,34].

Flavivirus structural proteins prM and E and non-structural protein NS1 are glycoproteins. Their glycosylation modifications play key roles in virus assembly and secretion. Tajima et al. found that DENV1 production was not detected when full-length DENV-1 RNA, which has an N-glycosylation site Asn130-to-Ala mutation in NS1, was transfected into cells [35]. In this study, the dengue virus with a deletion at amino acid residue 207 of the NS1 protein (N207-del) showed no detectable viral RNA and proteins during rescue and a lack of virus packaging after three successive passages. In addition, NS1 protein expression was significantly reduced in the N130 deletion mutant strain. Our results suggest that the N-glycosylation site of the DENV NS1 is essential for viral replication. These different observations may be attributed to differences in virus type, virus strain, and cells.

The mutation of the N-glycosylation site of the dengue virus NS1 protein to different amino acids may affect the replication and virulence of the virus. Based on previous findings [28,35,36,37,38,39], we designed various amino acid mutation strategies. Among the glycosylation site mutant viruses designed in this study, the neurovirulence of N130-del and N130del+207-209QQA strains to suckling mice was significantly reduced, especially N130del+207-209QQA, which showed markedly attenuated phenotype in both ICR and BALB/c suckling mice. However, the mortality of suckling mice infected intracerebrally with N130del+207-209QQA was higher at low doses than at high doses. The problem persisted after repeating the experiment. It is speculated that there are several viruses with different genetic mutations due to the impurity of the rescue virus. Therefore, the genetic stability of the N130del+207-209QQA strain was studied, and the results showed that this strain was genetically unstable and prone to nucleotide insertion mutations. Therefore, a plaque assay was performed to purify the mutant strains, and the plaque-purified virus was used to re-infect the suckling mice intracerebrally. Genome sequencing of the purified clones showed that clone #11-puri9 with significantly attenuated phenotype possessed an insertional mutation at amino acid residue 130, where the original nucleotide deletion mutation occurred while still maintaining the deletion of the glycosylation sites at positions 130 and 207 (Figure 7). This insertion stabilized the genome and kept the significantly attenuated phenotype, indicating it could be used as a new strategy for developing live attenuated vaccines.

Recent studies have shown that deletion of the N-glycosylation site in the NS1 protein of flaviviruses such as YFV, WNV, and ZIKV attenuates or abolishes their neurovirulence in mice [39,40,41]. Whiteman et al. showed that mutating the three consecutive amino acid residues from NTT to QQA at the first glycosylation site in the NS1 protein of WNV abolished viral virulence in mice [28]. Further studies revealed that the underlying mechanism is that the lack of glycosylation in the NS1 protein hinders viral replication and blocks NS1 protein maturation and secretion, leading to altered viral ultrastructure and subsequent attenuation of neurovirulence in mice [42]. Similarly, Pryor et al. showed that a mutation (N207A) of the glycosylation site at amino acid residue 207 in the NS1 protein in DENV-2 led to a significant reduction in virulence of dengue virus to 3-day-old BALB/c suckling mice when intracerebrally infecting the mice with 10 PFU of the virus [43]. However, our study found that the same mutation in DENV-4, (N207A) did not attenuate virulence, and the #11-puri9 strain (KN129-130TK) with additional glycosylation site deletion completely abolished virulence after intracranial infection to suckling mice at a dose of 100 PFU. These results suggest that deleting N-glycosylation sites in the NS1 protein provides a novel approach for developing genetically stable, attenuated dengue vaccines.

It is generally accepted that a dengue vaccine should induce strong and long-lasting specific neutralizing antibodies and cellular immune responses against all four dengue virus serotypes simultaneously to effectively prevent dengue hemorrhagic fever and dengue shock syndrome caused by infection with other serotypes in the endemic areas. Takeda Pharmaceuticals developed a tetravalent attenuated dengue chimeric vaccine using the PDK-53 strain as a vaccine candidate against DENV-2. Three critical attenuating sites with 5′UTR-57, NS1-53, and NS3-250 were identified in the non-structural proteins of the PDK-53 strain using reverse genetics techniques [44]. The chimeric virus vaccine candidates ChiDEN-2/1, ChiDEN-2/3, and ChiDEN-2/4 constructed using the PDK-53 strain as the backbone showed complete loss of neurovirulence against suckling mice with a promising safety profile [12]. Our previous studies showed that the rDENV4/1 construct remains strongly neurovirulent to suckling mice, indicating it is challenging to be used as a safe dengue vaccine candidate strain for further development. Therefore, we introduced a potential virulence locus identified in the DENV4-Ban18HK20 strain into the rDENV4/1 chimeric virus to further investigate its effect on virulence attenuation. The rDENV4/1 with this mutant locus showed a significantly attenuated phenotype in plaque size and neurovirulence in suckling mice, making it a promising candidate for further development of novel dengue chimeric vaccines.

Construction of infectious clones of flaviviruses and their chimeric viruses can be challenging. To overcome this, some researchers used in vitro ligation, where the full-length viral genome was first divided into several fragments, which were then ligated using DNA ligase. For each subclone, the fragments were digested using restriction enzymes (e.g., *Bsm*BI), reverse self-cleaved to prevent self-ligation, and subcloned. Subsequently, several linearized target fragments from subclones were ligated into a full-length viral cDNA in vitro using DNA ligase and used as a template for in vitro transcription and following viral rescue [45,46,47,48,49]. The method has been used to successfully construct larger SARS-CoV-2 recombinant virus and its reporter virus in a short period [50]. This study also used this approach and successfully constructed the recombinant chimeric dengue virus rDENV4/1. However, the viruses rescued through this approach generally have poor purity and need to be further purified using plaque assay to obtain genetically stable viruses. The approach can be applied to construct other serotypes of dengue chimeric viruses, laying the foundation for developing tetravalent dengue chimeric vaccines.

## 5. Conclusions

N-glycosylation modifications in the NS1 protein of flaviviruses are essential for their virulence. In this study, we identified mutant viruses with deletions in the N-glycosylation site of the NS1 protein, resulting in attenuated virulence in suckling mice. The finding highlights the importance of N-glycosylation modifications of the non-structural protein region in the virulence of flaviviruses. The attenuated virus could serve as a backbone for the construction of live attenuated dengue chimeric vaccines, providing a theoretical basis for the pathogenic mechanism of the dengue virus and the development of its potential live attenuated vaccines.

## Figures and Tables

**Figure 1 vaccines-11-00959-f001:**
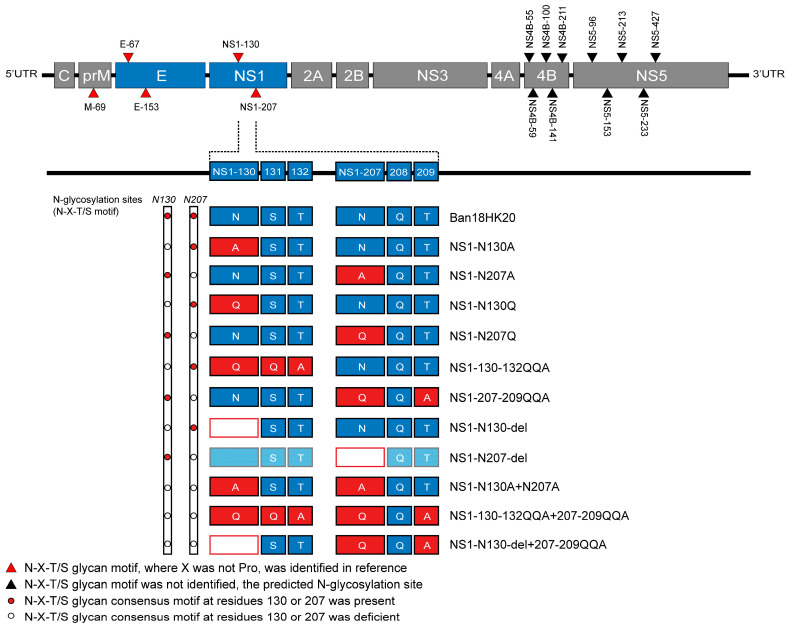
Construction of infectious clones with mutations at the N-glycosylation site in the NS1 proteins of the dengue virus. The letters in the blue or red boxes indicate amino acid residues at the N-linked glycosylation site of the NS1 protein. N, S, T, Q, and A indicate asparagine, serine, threonine, glutamine and alanine, respectively.

**Figure 2 vaccines-11-00959-f002:**
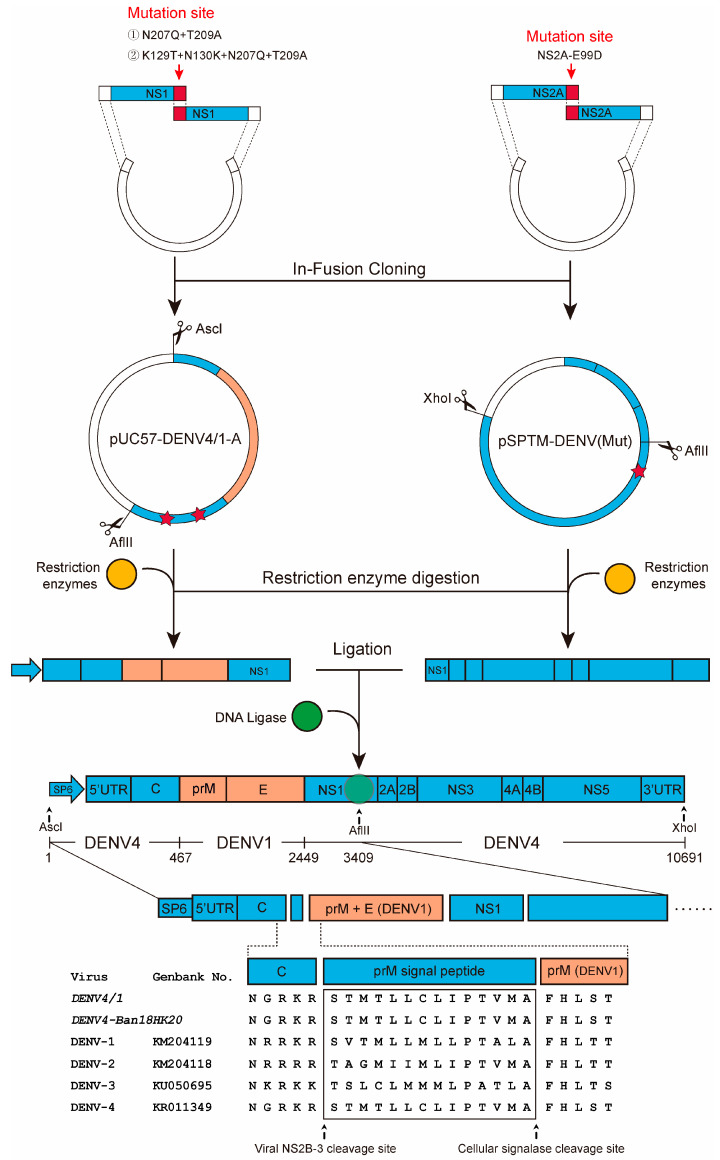
Construction of chimeric dengue virus DENV4/1 via in vitro ligation. The prM-E gene of DENV type 1 was cloned in the pUC57 vector as fragment A, and the backbone gene of DENV type 4 was used as fragment B. The two linear fragments, A and B, were then ligated in vitro using T4 DNA ligase to obtain the full-length viral cDNA. Finally, the chimeric virus was rescued by in vitro transcription and electro-transfection. The pentagrams in plasmid pUC57-DENV4/1-A indicate amino acid mutations at the N-glycosylation sites at positions 130 and 207, respectively, in the NS1 protein. The pentagram in plasmid pSPTM-DENV(Mut) indicates a mutation at amino acid 99 in the NS2A protein.

**Figure 3 vaccines-11-00959-f003:**
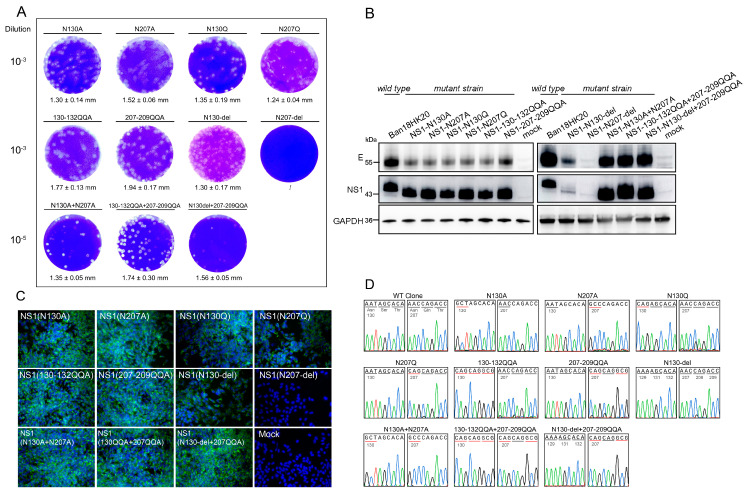
Identification of rescue viruses. (**A**) Plaque size of the dengue virus with mutations in the NS1 protein glycosylation site. The plaque size is indicated as average diameter of plaques ± standard deviation. (**B**) Expression of viral E and NS1 proteins in the parent strain of dengue virus and its NS1 protein glycosylation site mutant strain. (**C**) Indirect immunofluorescence assay of dengue virus parent strain and its NS1 protein glycosylation site mutant strain for specific E protein expression. (**D**) Genome sequencing of dengue virus and its mutant strains. The red lines below the bases indicate the mutation sites.

**Figure 4 vaccines-11-00959-f004:**
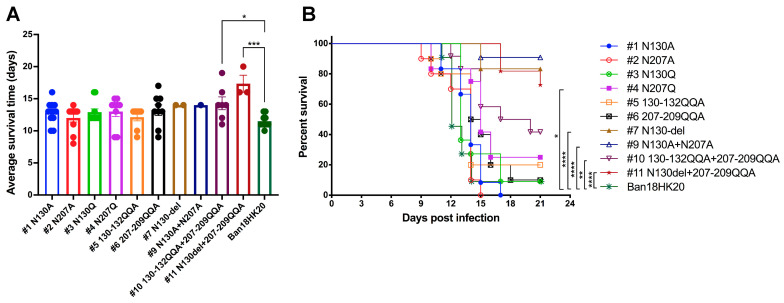
Neurovirulence of 100 PFU dengue virus mutated at the N-glycosylation site in the NS1 protein to 4-day-old ICR suckling mice. (**A**) Average survival time of mutant and parental viruses in 4-day-old ICR suckling mice inoculated with 100 PFU of mutant and parental viruses via the intracerebral route. * *p* ≤ 0.05, *** *p* ≤ 0.001. (**B**) Survival of 4-day-old ICR suckling mice inoculated with 100 PFU of mutant and parental viruses via the intracerebral route following 21 days of observation. Statistical significance was analyzed using Log-rank (Mantel–Cox) survival analysis. * *p* ≤ 0.05, ** *p* ≤ 0.01 and **** *p* ≤ 0.0001.

**Figure 5 vaccines-11-00959-f005:**
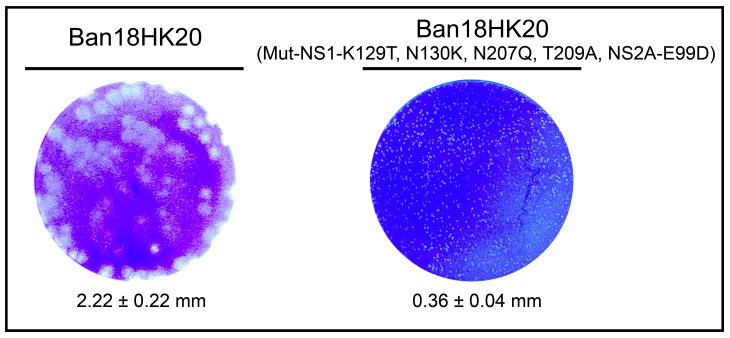
Plaque size of the plaque-purified attenuated rescued strain Ban18HK20 (Mut-NS1-K129T, N130K, N207Q, T209A, NS2A-E99D) and its non-mutated virus Ban18HK20. The plaque size is indicated as average diameter of plaques ± standard deviation.

**Figure 6 vaccines-11-00959-f006:**
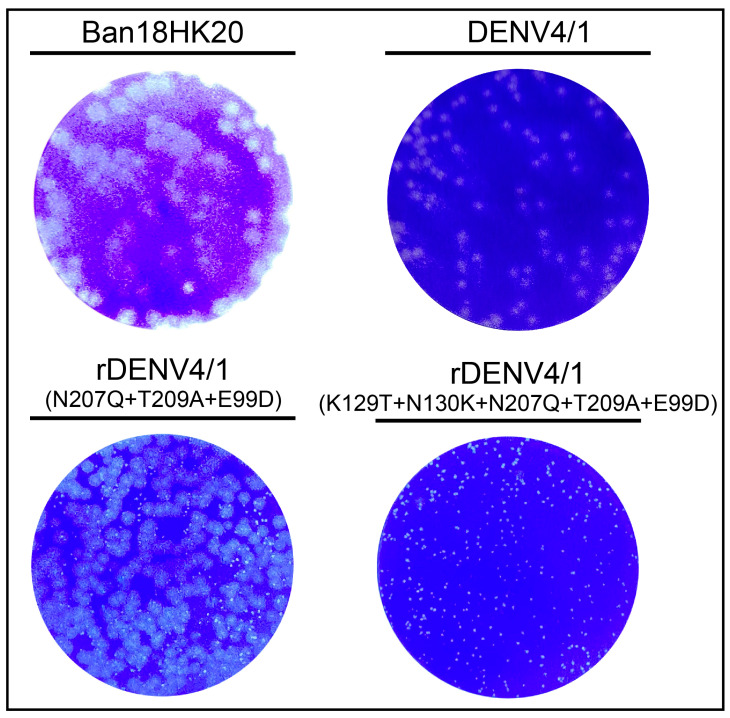
Plaque size of dengue chimeric viruses with and without amino acid mutations.

**Figure 7 vaccines-11-00959-f007:**
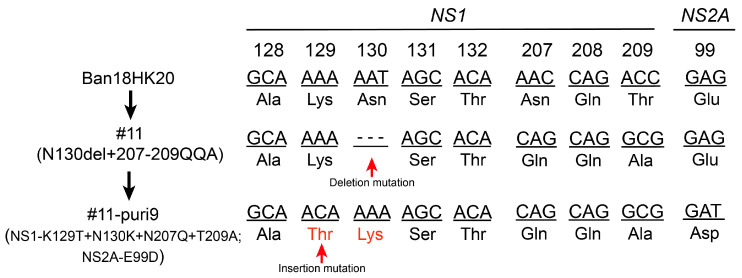
Screening of the attenuated dengue virus strains and associated virulence loci derived from adaptive insertional mutations following deletion mutations.

**Table 1 vaccines-11-00959-t001:** Neurovirulence of different dengue virus strains with mutations at the glycosylation sites of the NS1 protein in the 3-day-old BALB/c suckling mice.

Virus (Strain)	Dose (log_10_ PFU)	No. of Dead/Total (% Mortality)	LD_50_ (PFU)	AST ± SEM ^a^
Ban18HK20	2.2	6/6 (100)	1.20	9.2 ± 0.3
	1.2	3/4 (75)		12.0 ± 1.0
	0.2	4/6 (67)		14.5 ± 1.3
	−0.8	0/6 (0)		NA
#1 N130A	3.4	4/4 (100)	1.23	9.0 ± 0.4
	2.4	6/6 (100)		10.3 ± 0.9
	1.4	6/6 (100)		11.3 ± 1.0
	0.4	3/6 (50)		12.0 ± 0.6
#2 N207A	3.6	7/7 (100)	6.92	9.6 ± 0.4
	2.6	3/3 (100)		12.3 ± 0.3
	1.6	5/6 (83)		15.2 ± 1.6
	0.6	2/6 (33)		11.0 ± 2.0
#3 N130Q	3.2	5/5 (100)	2.09	9.6 ± 0.7
	2.2	6/6 (100)		10.3 ± 0.6
	1.2	6/6 (100)		11.3 ± 0.7
	0.2	2/6 (33)		15.5 ± 0.5
#4 N207Q	3.5	6/6 (100)	1.55	9.2 ± 0.4
	2.5	5/5 (100)		11.4 ± 0.8
	1.5	6/6 (100)		10.8 ± 0.5
	0.5	3/5 (60)		16.0 ± 0.6
	−0.5	1/6 (17)		16.0 ± 0.0
#5 130-132QQA	3.4	6/6 (100)	2.51	10.3 ± 0.5
	2.4	7/7 (100)		10.6 ± 0.6
	1.4	5/5 (100)		13.4 ± 0.7
	0.4	3/7 (43)		17.3 ± 1.5
#6 207-209QQA	3.6	6/6 (100)	12.59	11.2 ± 0.9
	2.6	4/4 (100)		9.3 ± 0.8
	1.6	6/6 (100)		11.7 ± 0.4
	0.6	0/4 (0)		NA
#7 N130-del	3.9	6/6 (100)	13.8	10.7 ± 0.8
	2.9	6/6 (100)		11.7 ± 0.6
	1.9	5/6 (83)		13.2 ± 0.6
	0.9	3/7 (43)		13.0 ± 0.0
#9 N130A+N207A	3.1	5/5 (100)	1.26	10.0 ± 0.3
	2.1	6/6 (100)		10.0 ± 0.3
	1.1	6/6 (100)		11.8 ± 0.3
	0.1	3/6 (50)		15.0 ± 2.1
#10 130-132QQA+207-209QQA	3.7	6/6 (100)	0.5	11.3 ± 0.3
	2.7	5/5 (100)		12.8 ± 0.2
	1.7	6/6 (100)		13.0 ± 0.4
	0.7	6/6 (100)		16.0 ± 1.0
	−0.3	3/6 (50)		14.7 ± 1.5
#11 N130del+207-209QQA	5.1	6/6 (100)	5754.40	11.3 ± 0.6
	4.1	2/6 (33)		16.5 ± 1.5
	3.1	1/6 (17)		12.0 ± 0.0
	2.1	3/6 (50)		16.5 ± 1.5
	1.1	0/6(0)		NA
PBS	NA	0/6(0)	NA	NA

^a^ Average survival time in days ± standard error of the mean (SEM). NA, not applicable.

**Table 2 vaccines-11-00959-t002:** Neurovirulence of different dengue virus strains with mutations at the glycosylation sites of the NS1 protein in the 4-day-old ICR suckling mice.

Virus (Strain)	Dose (PFU)	No. of Dead/Total (% Mortality)	LD_50_ (PFU)	AST ± SEM ^a^
Ban18HK20	100	10/11 (91)	4.90	11.5 ± 0.3
	10	8/11 (73)		11.8 ± 0.6
	1	1/10 (10)		15.0 ± 0.0
#1 N130A	100	12/12 (100)	3.16	12.8 ± 0.5
	10	10/11 (91)		14.2 ± 0.6
	1	1/10 (10)		14.0 ± 0.0
#2 N207A	100	10/10 (100)	6.17	12.0 ± 0.6
	10	7/11 (64)		15.0 ± 0.8
	1	0/9 (0)		NA
#3 N130Q	100	10/11 (91)	3.80	12.9 ± 0.5
	10	6/10 (60)		14.7 ± 0.4
	1	4/10 (40)		15.0 ± 0.6
#4 N207Q	100	9/12 (75)	13.80	13.0 ± 0.8
	10	4/11 (36)		14.5 ± 1.2
	1	4/11 (36)		18.5 ± 0.3
#5 130-132QQA	100	8/10 (80)	11.48	12.1 ± 0.6
	10	2/10 (20)		14.0 ± 0.0
	1	7/11 (64)		17.0 ± 0.5
#6 207-209QQA	100	9/10 (90)	5.37	13.2 ± 0.8
	10	5/10 (50)		15.4 ± 0.9
	1	4/11 (36)		18.3 ± 0.3
#7 N130-del	100	2/12 (17)	>100	14.0 ± 0.0
	10	5/12 (42)		15.2 ± 0.5
	1	1/11 (9)		16.0 ± 0.0
#9 N130A+N207A	100	1/11 (9)	67.61	14.0 ± 0.0
	10	5/11 (45)		14.0 ± 0.6
	1	5/13 (38)		16.0 ± 0.6
#10 130-132QQA+207-209QQA	100	7/12 (58)	20.89	14.3 ± 1.0
	10	5/11 (45)		16.6 ± 0.9
	1	2/11 (18)		17.0 ± 1.0
#11 N130del+207-209QQA	100	3/11 (27)	100	17.3 ± 1.3
	10	3/12 (25)		19.3 ± 0.3
	1	2/11 (18)		18.0 ± 1.0
PBS	NA	0/12 (0)	NA	NA

^a^ Average survival time in days ± standard error of the mean (SEM). NA, not applicable.

**Table 3 vaccines-11-00959-t003:** Genetic stability of a dengue virus strain (#11) with mutations at the NS1 glycosylation sites.

Locus/Protein	Virus Nucleotide Changes (Amino Acid Changes)				
	Ban18HK20	#11-P0	#11-P1	#11-P3	#11-P5
NS1-112	AAA(K)	AAA(K)	AAA(K)	AAC(N)	AAC(N)
NS1-129	AAA(K)	AAA(K)	AAA(K)	ACA(T)	ACA(T)
NS1-130	AAT(N)	del	del	AAA(K)	AAA(K)
NS1-207	AAC(N)	CAG(Q)	CAG(Q)	CAG(Q)	CAG(Q)
NS1-209	ACC(T)	GCG(A)	GCG(A)	GCG(A)	GCG(A)

**Table 4 vaccines-11-00959-t004:** Neurovirulence of a plaque-purified dengue virus strain (#11) with mutations at the NS1 glycosylation in the 4-day-old ICR suckling mice.

Virus (Strain)	Dose (PFU)	No. of Dead/Total (% Mortality)	LD_50_ (PFU)	AST ± SEM ^a^
#11-puri1	100	1/6 (17)	>100	13.0 ± 0.0
	10	1/6 (17)		19.0 ± 0.0
	1	0/6 (0)		NA
#11-puri2	100	4/7 (57)	75.86	15.0 ± 1.1
	10	0/5 (0)		NA
	1	0/5 (0)		NA
#11-puri3	100	2/6 (33)	>100	17.5 ± 2.5
	10	1/7 (14)		19.0 ± 0.0
	1	0/6 (0)		NA
#11-puri4	100	0/5 (0)	>100	NA
	10	1/5 (20)		18.0 ± 0.0
	1	0/5 (0)		NA
#11-puri5	100	1/5 (20)	>100	19.0 ± 0.0
	10	1/5 (20)		17.0 ± 0.0
	1	0/5 (0)		NA
#11-puri6	100	1/6 (17)	>100	20.0 ± 0.0
	10	1/6 (17)		20.0 ± 0.0
	1	0/7 (0)		NA
#11-puri7	100	0/6 (0)	>100	NA
	10	0/5 (0)		NA
	1	0/6 (0)		NA
#11-puri8	100	4/6 (67)	56.23	17.8 ± 0.8
	10	0/6 (0)		NA
	1	0/6 (0)		NA
#11-puri9	100	0/5 (0)	>100	NA
	10	0/6 (0)		NA
	1	0/5 (0)		NA
#11-puri10	100	0/13(0)	>100	NA
	10	2/12(17)		19.5 ± 0.5
	1	1/11(9)		19.0 ± 0.0
PBS	NA	0/6 (0)	NA	NA

^a^ Average survival time in days ± standard error of the mean (SEM). NA, not applicable.

**Table 5 vaccines-11-00959-t005:** RT-PCR sequencing of the plaque-purified dengue virus strain (#11) with mutations at the NS1 glycosylation sites.

Locus/Protein	Virus Nucleotide Changes (Amino Acid Changes)											
	Ban18HK20	#11	#11-Puri1	#11-Puri2	#11-Puri3	#11-Puri4	#11-Puri5	#11-Puri6	#11-Puri7	#11-Puri8	#11-Puri9	#11-Puri10
NS1-112	AAA(K)	···	AAC(N)	···	···	···	···	···	···	···	···	AAC(N)
NS1-129	AAA(K)	···	···	ACA(T)	ACA(T)	ACA(T)	ACA(T)	ACA(T)	ACA(T)	ACA(T)	ACA(T)	···
NS1-130	AAT(N)	del	del	AAA(K)	AAA(K)	AAA(K)	AAA(K)	AAA(K)	AAA(K)	AAA(K)	AAA(K)	del
NS1-207	AAC(N)	CAG(Q)	CAG(Q)	CAG(Q)	CAG(Q)	CAG(Q)	CAG(Q)	CAG(Q)	CAG(Q)	CAG(Q)	CAG(Q)	CAG(Q)
NS1-209	ACC(T)	GCG(A)	GCG(A)	GCG(A)	GCG(A)	GCG(A)	GCG(A)	GCG(A)	GCG(A)	GCG(A)	GCG(A)	GCG(A)
NS2A-99	GAG(E)	···	GAT(D)	···	···	···	···	···	GAT(D)	···	GAT(D)	···
NS3-4	CTG(L)	···	CTA(L)	···	···	···	···	···	···	···	···	···
NS4A-20	AGG(R)	···	AGA(R)	···	···	···	···	···	···	···	···	AGA(R)
NS4B-197	CCA(P)	···	CCT(P)	CCT(P)	CCT(P)	CCT(P)	CCT(P)	CCT(P)	CCT(P)	CCT(P)	CCT(P)	···
NS5-362	AGA(R)	···	···	CGA(R)	···	CGA(R)	CGA(R)	CGA(R)	CGA(R)	CGA(R)	CGA(R)	···
NS5-699	AAG(K)	···	···	···	AAT(N)	···	···	···	···	···	···	···
NS5-827	GAC(D)	···	···	···	GAT(D)	···	···	···	···	···	···	···

**Table 6 vaccines-11-00959-t006:** Neurovirulence of the rescued dengue virus mutants at the #11-puri9 virulence locus in the 4-day-old ICR suckling mice.

Virus (Strain)	Dose (PFU)	No. of Dead/Total (% Mortality)	LD_50_ (PFU)	AST ± SEM ^a^
Ban18HK20 (Mut-NS1-K129T, N130K, N207Q, T209A, NS2A-E99D)	100	0/12 (0)	>100	NA
	10	0/12 (0)		NA
	1	0/13 (0)		NA
Ban18HK20	100	12/12 (100)	19.05	10.5 ± 0.3
	10	3/12 (25)		15.3 ± 0.9
	1	1/13 (8)		14.0 ± 0.0
PBS	NA	0/12 (0)	NA	NA

^a^ Average survival time in days ± standard error of the mean (SEM). NA, not applicable.

**Table 7 vaccines-11-00959-t007:** Neurovirulence of chimeric dengue virus DENV4/1 and its mutants in the 4-day-old ICR suckling mice.

Virus (Strain)	Dose (PFU)	No. of Dead/Total (% Mortality)	LD_50_ (PFU)	AST ± SEM ^a^
rDENV4/1(N207Q+T209A+E99D)	100	14/14 (100)	2.24	9.4 ± 0.2
	10	8/11 (73)		11.3 ± 0.4
	1	5/12 (42)		11.8 ± 0.4
rDENV4/1(K129T+N130K+N207Q+T209A+E99D)	100	0/19 (0)	>100	NA
	10	0/15 (0)		NA
	1	0/20 (0)		NA
DENV4/1	100	15/15 (100)	2.51	8.7 ± 0.2
	10	15/16 (94)		10.1 ± 0.2
	1	3/14 (21)		10.7 ± 0.3
PBS	NA	0/12 (0)	NA	NA

^a^ Average survival time in days ± standard error of the mean (SEM). NA, not applicable.

**Table 8 vaccines-11-00959-t008:** Genetic stability of the purified and rescued dengue virus #11-puri9 strains.

Locus/Protein	Virus Nucleotide Changes (Amino Acid Changes)						
	Ban18HK20	#11	#11-Puri9	#11-Puri9-P3	#11-Puri9-P5	#11-Puri9 (Rescued)-P3	#11-Puri9 (Rescued)-P5
NS1-129	AAA(K)	···	ACA(T)	ACA(T)	ACA(T)	ACA(T)	ACA(T)
NS1-130	AAT(N)	del	AAA(K)	AAA(K)	AAA(K)	AAA(K)	AAA(K)
NS1-207	AAC(N)	CAG(Q)	CAG(Q)	CAG(Q)	CAG(Q)	CAG(Q)	CAG(Q)
NS1-209	ACC(T)	GCG(A)	GCG(A)	GCG(A)	GCG(A)	GCG(A)	GCG(A)
NS2A-99	GAG(E)	···	GAT(D)	GAT(D)	GAT(D)	GAT(D)	GAT(D)
NS4B-197	CCA(P)	···	CCT(P)	CCT(P)	CCT(P)	CCA(P)	CCA(P)
NS5-362	AGA(R)	···	CGA(R)	CGA(R)	CGA(R)	AGA(R)	AGA(R)

## Data Availability

All data are available from the corresponding authors upon request.

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
