# Peer review of "NS1 Protein N-Linked Glycosylation Site Affects the Virulence and Pathogenesis of Dengue Virus"

_vaccines, 2023, doi:10.3390/vaccines11050959_

Round 1

Reviewer 1 Report

The manuscript describes development of the attenuated dengue virus strain using site-directed mutagenesis in NS1 and NS2A proteins. The results show that the deletion of selected amino acids in the NS1 protein affected its packaging and stability.

My only comment is to add two references for the study. Tajima et al., 2008. Virus Genes 36:323-329 and Wang et al., 2019. PLoS Pathog. 15:e1007938. 

Author Response

Reviewer: 1

Q1:

My only comment is to add two references for the study. Tajima et al., 2008. Virus Genes 36:323-329 and Wang et al., 2019. PLoS Pathog. 15:e1007938.

R1:

Thank you very much for your careful review and suggestions, we have added citations to these two papers in our discussion.

Reviewer 2 Report

In this manuscript, Fang and others described the effect of DENV NS1 glycosylation mutations on viral pathogenesis. Among 11 NS1 mutants, a genetically stable attenuated strain #11-puri9 bearing four mutations on its parental strain N130del+207-209QQA was isolated as a neurovirulence-attenuated strain. The genetically stable mutant isolated by small plaque purification had five additional amino acid changes in the NS1 and NS2A proteins. The authors also generated DENV chimeric viruses by introducing these five mutated amino residues or three of them in an infection clone for a primary hamster kidney cell-culture-adapted strain of DENV-4 Ban18 to demonstrate that the two culture adapted mutations K129T and N130K were associated with the attenuation feature.  

Overall, this paper describes various degrees of virulence for the attenuated strains that they generated by mutagenesis and/or isolated via plaque purification. While the data were interesting in regards to application aspects of isolated attenuated strains, additional information needs to be shown for the authors to claim that some of the attenuated strains can indeed be used as live-attenuated vaccine candidates. Some parts of the manuscript were not clearly written and detailed methods are missing too.

Major issues

1. In line 110, was the IVT RNA capped? If so, describe in detail how cap was added at the 5’-end of the viral genome.

2. In Table 1, they provide % mortality of the tested mutants. What was the criteria of mortality in the animal experiments? The authors should show weight changes in suckling mice infected with mutant ZIKV stains (also for Figure 4B too). More critically, viral loads in brain tissues and sera would be informative to tell if the mutants had altered neurovirulence and replication competence in vivo. If IHC was done, please show the results too.

3. It appears that the NS1 mutants tested in vitro and in vivo were all attenuated in their virulence and replication capacity with varying degrees. Despite these phenotype changes, there is no direct evidence that the attenuated features were caused by changes in NS1 glycosylation. The authors may examine glycosylation status of the NS1 mutants and analyze if their secretion was altered.

  Regarding to this issue, a mutant with N130A and N207A expressed a lower level of E protein (in Figure 3Bl while NS1 expression was enhanced) and displayed a substantially attenuated feature in ICR suckling mice (Figure 4B). I was wondering if the attenuation feature was caused by the decreased E expression. It is important to show that the two mutations K129T and N130K acquired during passages of one of isolated attenuated strains caused any changes in NS1 glycosylation.

4. In Figure 3A, representative images of plaques formed by the rescued viruses were presented. The authors mentioned that plaque sizes of the NS1 mutants were not “significantly different”. Please show the data showing the average diameter of plaques and their relative sizes in comparison to that of parental virus. In Figure 3B, what was the MOI of infection for each rescued viruses? Was it possible to grow N207-del mutant? It appears to be defective in propagation since no plaque was formed in the panel A and no NS1 was detectable by immunostaining. Was the virus used in A and B panels P1 stocks (passaged once following recovering P0 that was collected after IVT transfection)?

5. What are the red and open circles presented below the N-glycan mark in Figure 2? Red circles seem to be glycosylation positive sites. Any experimental evidence? If not, the symbols would be predicted phenotypes.

6. Discuss further what might the possible reasons for the failure of rescue of recombinant N207-del?

7. Plaque sizes in diameter can be shown along with the representative images in Figure 5. The authors also have to show growth kinetics of #11-puri9 and its parental recombinant strains (N130del+207-209QQA and Ban18HK20).

8. In section 3.4., the N130del+207-209QQA was subjected to plaque purification to isolate ten clonal variants. However, in Table 6, one of the purified plaque #11-puri9 was presented as a derivative of Ban18HK20. This has to be clarified! What was the parental strain of #11-puri9?

9. What was the rationale for the construction of rDENV4/1 with K129T+N130K+N207Q+T209A+E99D mutations?

10. In the method section, RNA sequencing method, depth of reads, bioinformatics tools used to analyze viral RNA quasispecies have not been described at all. 

Minor points

1. In lines 227 and 228, lg should read log10.

2. In line 262, “lower LD50” must read “greater LD50 (5754.40)”?

Author Response

Responses to reviewer

Reviewer: 2

Major issues

Q1:

In line 110, was the IVT RNA capped? If so, describe in detail how cap was added at the 5’-end of the viral genome.

R1:

We appreciate your careful review. Since the 5' end of native dengue virus RNA has a cap structure, we have added cap analogues during in vitro transcription. Relevant descriptions have been supplemented to the methods of the paper.

Q2:

In Table 1, they provide % mortality of the tested mutants. What was the criteria of mortality in the animal experiments? The authors should show weight changes in suckling mice infected with mutant ZIKV stains (also for Figure 4B too). More critically, viral loads in brain tissues and sera would be informative to tell if the mutants had altered neurovirulence and replication competence in vivo. If IHC was done, please show the results too.

R2:

Thanks for your careful review. Because dengue virus is highly sensitive to suckling mice following an intracerebral challenge, mortality was determined by daily observation after the challenge until the suckling mice became ill and died, and then one was counted as a fatality. The ratio of the number of dead mice to the total number of mice was counted as the mortality rate of suckling mice. The mortality results were not statistically analyzed by us, but for some mutated strains, such as #11 N130del+207-209QQA, significantly reduced or no mortality was observed in suckling mice challenged with the same dose of virus. Our current results did not record and detect the following experiments, such as weight changes in suckling mice, viral load in brain tissue and serum, and IHC assays. These more detailed studies will be further examined.

However, in Table 1, we evaluated the virulence of different mutant viruses by three indicators: mortality, LD50 and average survival time of mice after challenged. In Table 2, different mice were used to verify the virulence of the mutant viruses. And in Figure 4, the data on the average survival time and survival curve of different mutant virus strains were statistically analyzed. The above assays were combined to evaluate the virulence of the mutant viruses compared to the parent strains, in order to identify the strains with significantly reduced virulence for further study.

Q3:

It appears that the NS1 mutants tested in vitro and in vivo were all attenuated in their virulence and replication capacity with varying degrees. Despite these phenotype changes, there is no direct evidence that the attenuated features were caused by changes in NS1 glycosylation. The authors may examine glycosylation status of the NS1 mutants and analyze if their secretion was altered.

Regarding to this issue, a mutant with N130A and N207A expressed a lower level of E protein (in Figure 3Bl while NS1 expression was enhanced) and displayed a substantially attenuated feature in ICR suckling mice (Figure 4B). I was wondering if the attenuation feature was caused by the decreased E expression. It is important to show that the two mutations K129T and N130K acquired during passages of one of isolated attenuated strains caused any changes in NS1 glycosylation.

R3:

Thank you very much for your suggestion. We did not perform the identification of the glycosylation status of the NS1 protein from the mutated virus in this study, but only performed the mutation at the gene level for the consensus glycosylation sites and the virulence of the mutated virus. In our later studies, we will conduct SDS-PAGE electrophoresis of the mutant virus to obtain NS1 protein bands and to detect the glycosylation level and glycoform type of the mutated NS1 protein by mass spectrometry. On the other hand, glycosylation modifications of mutated viruses will also be investigated by digestion with glycosidases. For the two mutations K129T and N130K acquired during passages, which altered the consensus glycosylation site at N130 (N-X-S/T), but we also did not validate the glycosylation modification of the viral NS1 protein, which will also be the next step in our study. However, our current study focuses more on the affect of virulence at the viral gene level by altering the glycosylation site.

Q4:

In Figure 3A, representative images of plaques formed by the rescued viruses were presented. The authors mentioned that plaque sizes of the NS1 mutants were not “significantly different”. Please show the data showing the average diameter of plaques and their relative sizes in comparison to that of parental virus. In Figure 3B, what was the MOI of infection for each rescued viruses? Was it possible to grow N207-del mutant? It appears to be defective in propagation since no plaque was formed in the panel A and no NS1 was detectable by immunostaining. Was the virus used in A and B panels P1 stocks (passaged once following recovering P0 that was collected after IVT transfection)?

R4:

Thanks for your suggestions. We have added the average diameter of plaques in Figure 3A. In Figure 3B, we used a rescue virus with an MOI of 1 to infect the cells as described in Materials and Methods (2.6 Western blotting). For N207-del mutant virus, we used both P0 generation virus for plaque and Western blotting assays, and no virus was detected by either way. We also extracted RNA from the P0 generation virus to perform RT-PCR amplification, and no virus was detected. To avoid the possibility of undetectable primary rescue virus by low viral titer, we also passaged N207-del mutant virus of P0 generation on Vero cells for 3 generations for virus amplification, and the amplified virus detection results showed that no virus was detected either. We also performed two independent parallel experiments for the rescue of the N207-del mutant virus, and both results also showed no detection of the virus. Therefore, we conclude that a deletion mutation in amino acid 207 of the NS1 protein in DENV4 Ban18HK20 strain may directly affect virus replication or packaging.

Q5:

What are the red and open circles presented below the N-glycan mark in Figure 2? Red circles seem to be glycosylation positive sites. Any experimental evidence? If not, the symbols would be predicted phenotypes.

R5:

Thanks for your suggestions. The red and open circles represent the status of the N-glycosylation sites. The relevant notes in Figure2 have been added, so please review them again.

Q6:

Discuss further what might the possible reasons for the failure of rescue of recombinant N207-del?

R6:

Thanks for your suggestion and we have further discussed the reasons in the second paragraph of the discussion. It has been reported that missense mutations in the N glycosylation site of the NS1 protein from different types of dengue viruses caused the virus fail to replicate. Although the mutation sites were different, they also resulted in virus rescue failure, and we have passaged the supernatant of the failed virus in Vero cells for 3 successive generations, and no virus was detected in any of them. In addition, we also performed independent replicate experiments of N207-del virus rescue, all of which yielded the same results. Therefore, it was demonstrated that amino acid N207 of NS1 protein in this study is necessary for viral replication of DENV-4 Ban18HK20 strain.

Q7:

Plaque sizes in diameter can be shown along with the representative images in Figure 5. The authors also have to show growth kinetics of #11-puri9 and its parental recombinant strains (N130del+207-209QQA and Ban18HK20).

R7:

Thank you for your review and suggestions. We have added plaque sizes in diameter in Figure 5. We did not perform growth kinetics of #11-puri9 and its parental recombinant strains, but we also performed Sanger sequencing of the viral genome for the rescue virus and identified that the sequences were all correct.

Q8:

In section 3.4., the N130del+207-209QQA was subjected to plaque purification to isolate ten clonal variants. However, in Table 6, one of the purified plaque #11-puri9 was presented as a derivative of Ban18HK20. This has to be clarified! What was the parental strain of #11-puri9?

R8:

In section 3.4, we screened 10 clones of purified strains of virus #11-puri9 that were not lethal to suckling mice challenged with 3 doses (100 PFU, 10 PFU, 1 PFU) (see Table 4). By viral genome sequencing, five amino acid loci were identified to be mutated (NS1-K129T, N130K, N207Q, T209A, and NS2A-E99D) compared to Ban18HK20 virus (see Table 5). To verify whether these five amino acid site mutations were responsible for the virulence reduction, we introduced these five mutant sites into the primary Ban18HK20 virus for virus rescue by constructing infectious clones. Rescue virus Ban18HK20 (Mut-NS1-K129T, N130K, N207Q, T209A, NS2A-E99D) and the primary non-mutated virus Ban18HK20 were used for suckling mice neurovirulence assays (see Table 6 for results). In Table 6, "Ban18HK20 (Mut-NS1-K129T, N130K, N207Q, T209A, NS2A-E99D)" is the rescue virus that contains these 5 amino acid mutations with the same neurovirulence against suckling mice as the purified virus strain #11-puri9 (all non-lethal to suckling mice challenged with 100 PFU). While the results of neurovirulence for purified virus strain #11-puri9 against suckling mice were shown in Table 4.

Q9:

What was the rationale for the construction of rDENV4/1 with K129T+N130K+N207Q+T209A+E99D mutations?

R9:

Thank you for reviewing. We constructed this chimeric virus for two reasons. On the one hand, to investigate whether the five amino acid mutation sites we identified are also characterized by small plaques and low neurovirulence in chimeric viruses. To provide a theoretical basis for the subsequent construction of attenuated dengue chimeric viruses for other serotypes. On the other hand, our previously constructed rDENV4/1 chimeric virus is highly neurovirulent, so we are trying to reduce the viral neurovirulence by introducing these mutations which would lay the foundation for the development of a tetravalent attenuated dengue chimeric vaccine.

Q10:

In the method section, RNA sequencing method, depth of reads, bioinformatics tools used to analyze viral RNA quasispecies have not been described at all.

R10:

Thank you, the amplification products of this study were sent to Sangon Biotech Co., Ltd for Sanger sequencing, and the relevant descriptions in the paper have been revised.

Minor points

Q1:

In lines 227 and 228, lg should read log10.

R1:

Thank you very much for your careful review, it has been revised.

Q2:

In line 262, “lower LD50” must read “greater LD50 (5754.40)”?

R2:

Sorry for the mistake here, corrected.

Reviewer 3 Report

The authors used reverse genetics of dengue virus (DENV) to introduce mutations at the N-glycosylation site of NS1 to generate recombinant viruses without glycosylation. As a result of analyzing the generated recombinant virus, they clarified that NS1 glycosylation is involved in DENV virulence and pathogenesis. In addition, some viruses among them were genetically stable, and were shown to be candidates for DENV vaccines. These results show that the reverse genetics method can be used for the development of novel DENV vaccines, which is a very interesting study.

[Major points]

1. The primers used in the PCR and sequencing should be described in a little more detail. In particular, the differences between F1 and F2, R1, and R2 in Table S1 also need explanation.

2. Result 3.1. might be described in the Methods.

[Minor points]

1. P3, line 12, line 10 from the top: Describe the composition of "RIPA".

2. Figure 2. Describe what the red and black arrows on the DENV genome above represent. Also, I think the authors need to explain what the two bars on the far left in the diagram below, which shows the NS1 mutation site, represent.

3. P12, line 31, line 5 from the top: "attenuated viruses had smaller plaques" needs references.

4. Figure 6; In the figure of rDENV4/1 (N207Q+T209A+E99D), small and large plaques seem to coexist. Is this correct?

5. Figure 7: Is the red “Lys” in #11-puri9 (129-130-KN/TK) also an “insertion mutation”? If so, there should also be a haze arrow under the red "Lys".

6. P13, line 38, 4th line from top: Figure 7 shows only the 130th mutation in the amino acid sequence of NS1. How about inserting the 207 s as well?

Author Response

Responses to reviewer 3

Reviewer: 3

Major points

Q1:

The primers used in the PCR and sequencing should be described in a little more detail. In particular, the differences between F1 and F2, R1, and R2 in Table S1 also need explanation.

R1:

Thank you for your suggestions. The sequencing primers have been supplemented to Table S2. The primers for clone construction in Table S1 are described in "2.2. Construction strategy for molecular cloning" and Figure1 (clone construction schematic). Briefly, two pairs of primers are used to amplify two fragments containing homologous sequences (e.g. F1 and N130A-R1 to amplify fragment 1, and N130A-F2 and R2 to amplify fragment 2) for subsequent homologous recombination cloning with vector fragments.

Q2:

Result 3.1. might be described in the Methods.

R2:

Thank you very much for your review, the method and results have been revised.

Minor points

Q1:

P3, line 12, line 10 from the top: Describe the composition of "RIPA".

R1:

Thank you, the description has been added.

Q2:

Figure 2. Describe what the red and black arrows on the DENV genome above represent. Also, I think the authors need to explain what the two bars on the far left in the diagram below, which shows the NS1 mutation site, represent.

R2:

Thank you very much for your review suggestion. The relevant notes in Figure 2 have been added.

Q3:

P12, line 31, line 5 from the top: "attenuated viruses had smaller plaques" needs references.

R3:

Thank you very much for your careful review, we have cited two references in the paper.

Q4:

Figure 6; In the figure of rDENV4/1 (N207Q+T209A+E99D), small and large plaques seem to coexist. Is this correct?

R4:

Thanks for your review. The attenuated virus obtained by plaque purification has five amino acid mutations (K129T, N130K, N207Q, T209A, E99D). To verify whether all five mutation sites affect the virulence and genetic stability of the attenuated viruses obtained by purification, rescue viruses with five mutation sites and rescue viruses with only three of the five mutation sites were constructed. The results showed that viruses containing five amino acid mutations had the same small plaques as the purified strain, while viruses containing only three amino acid mutations showed the co-existence of large and small plaques, which also suggested that viruses containing only three amino acid mutations (N207Q+T209A+E99D) might be genetically unstable and that viruses were prone to mutations with the characteristic of co-existence of large and small plaques. We also conducted parallel control experiments and found that the coexistence of large and small plaques was not caused by experimental manipulation, but possibly by the instability of the virus strain caused by the three mutant loci. We believe that this result is reasonable and explainable.

Q5:

Figure 7: Is the red “Lys” in #11-puri9 (129-130-KN/TK) also an “insertion mutation”? If so, there should also be a haze arrow under the red "Lys".

R5:

Thank you for reviewing this. The red "Lys" is not an insertion mutation relative to strain #11 (N130del+207-209QQA), but rather a frame shift mutation caused by an insertion mutation at amino acid site 129. However, for strain Ban18HK20, missense mutations occurred in amino acids 129 and 130 of strain #11-puri9 (129-130-KN/TK). So I don't think we need to add arrows here.

Q6:

P13, line 38, 4th line from top: Figure 7 shows only the 130th mutation in the amino acid sequence of NS1. How about inserting the 207 s as well ?

R6:

Thank you very much for your careful review, and Figure 7 has been revised.

Round 2

Reviewer 2 Report

The authors have addressed most of the comment from this reviewer to improve the manuscript.